# Oblique and Asymmetric Klein Tunneling across Smooth NP Junctions or NPN Junctions in 8-*Pmmn* Borophene

**DOI:** 10.3390/nano11061462

**Published:** 2021-05-31

**Authors:** Zhan Kong, Jian Li, Yi Zhang, Shu-Hui Zhang, Jia-Ji Zhu

**Affiliations:** 1School of Science and Laboratory of Quantum Information Technology, Chongqing University of Posts and Telecommunications, Chongqing 400065, China; kongz2021@163.com (Z.K.); jianli@cqupt.edu.cn (J.L.); zhangyia@cqupt.edu.cn (Y.Z.); 2College of Mathematics and Physics, Beijing University of Chemical Technology, Beijing 100029, China

**Keywords:** Klein tunneling, borophene, Dirac fermions

## Abstract

The tunneling of electrons and holes in quantum structures plays a crucial role in studying the transport properties of materials and the related devices. 8-Pmmn borophene is a new two-dimensional Dirac material that hosts tilted Dirac cone and chiral, anisotropic massless Dirac fermions. We adopt the transfer matrix method to investigate the Klein tunneling of massless fermions across the smooth NP junctions and NPN junctions of 8-Pmmn borophene. Like the sharp NP junctions of 8-Pmmn borophene, the tilted Dirac cones induce the oblique Klein tunneling. The angle of perfect transmission to the normal incidence is $20.4^{\circ }$, a constant determined by the Hamiltonian of 8-Pmmn borophene. For the NPN junction, there are branches of the Klein tunneling in the phase diagram. We find that the asymmetric Klein tunneling is induced by the chirality and anisotropy of the carriers. Furthermore, we show the oscillation of electrical resistance related to the Klein tunneling in the NPN junctions. One may analyze the pattern of electrical resistance and verify the existence of asymmetric Klein tunneling experimentally.

## 1. Introduction

Two-dimensional (2D) materials have been the superstars for their novel properties in condensed matter physics since its first isolation of graphene in 2004 [1]. Right now, the booming 2D materials family includes not just graphene and the derivatives of graphene but also transition metal dichalcogenides (TMDs) [2,3,4], black phosphorus [5,6,7,8], indium selenide [9,10,11], stanene [12,13], and many other layered materials [14,15]. Among these 2D materials, the so-called Dirac materials host massless Dirac fermions, always in the spotlight. Carriers in 2D Dirac materials usually have chirality or pseudospin from two atomic sublattices. Together with chirality, the linear Dirac dispersion gives rise to remarkable transport properties, including the absence of backscattering [1,16,17]. Due to the suppression of backscattering, massless Dirac fermions could tunnel a single square barrier with 100% transmission probability. This surprising result has been known as Klein tunneling [16,18,19,20,21]. Klein tunneling is the basic electrical conduction mechanism through the interface between *p*-doped and *n*-doped regions. Klein tunneling’s elucidation plays a key role in designing and inventing electronic devices based on 2D Dirac materials.

Recently, several 2D boron structures have been predicted and experimentally fabricated [22,23,24,25]. The 8-Pmmn borophene belongs to the space group Pmmn, which means an orthorhombic lattice has an mmm symmetric point group (three-mirror symmetry planes perpendicular to each other) combine with a glide plane at one of the mirror symmetry planes [22,26]. This kind of structure is the most stable symmetric phase of borophene and may be kinetically stable at ambient conditions. It revealed the tilted Dirac cone and anisotropic massless Dirac fermions by first-principles calculations [27,28].These unique Dirac fermions attracts people to explore the various physical properties such as strain-induced pseudomagnetic field [29], anisotropic density–density response [30,31,32,33], optical conductivity [34,35], modified Weiss oscillation [36,37], borophane and its tight-binding model [37], nonlinear optical polarization rotation [38], oblique Klein tunneling [39,40,41], few-layer borophene [42,43], intense light response [44,45], RKKY interaction [46,47], anomalous caustics [48], electron–phonon coupling [49], valley–contrast behaviors [50,51], Andreev reflection [52], and so on. The oblique Klein tunneling, the deviation of the perfect transmission direction to the normal direction of the interface, is induced by the anisotropic massless Dirac fermions or the tilted Dirac cone [39,53]. However, the on-site disorder or smoothing of the NP junction interface or the square potential may destroy the ideal Klein tunneling, which means the sharp interface strongly depends on high-quality fabrication state-of-the-art technology [54]. Therefore, the detailed discussion of the smooth NP junction and the tunable trapezoid potential would be helpful for the promising electronic devices based on 2D Dirac materials.

In this paper, we study the transmission properties of anisotropic and tilted massless Dirac fermions across smooth NP junctions and NPN junctions in 8-Pmmn borophene. Similar to the sharp NP junction, the oblique Klein tunneling retains due to the tilted Dirac cone. This conclusion does not depend on the NP junctions’ doping levels as the normal Klein tunneling but depends on the junction direction. We show the angle of oblique Klein tunneling is $20.4^{\circ }$, a constant determined by the Hamiltonian parameters of 8-Pmmn borophene. For the NPN junction, there are branches of the Klein tunneling in the phase diagram. We find that the asymmetric Klein tunneling is induced by the chirality and anisotropy of the carriers [55]. The indirect consequence of the asymmetric Klein tunneling lies in the oscillation of the electrical resistance. The analysis of the pattern of the oscillation of electrical resistance would help verify the existence of asymmetric Klein tunneling experimentally.

The rest of the paper is organized as follows. In Section 2, we introduce the Hamiltonian and the energy spectrum for the 8-Pmmn borophene, the NP and NPN junction’s potential, and present the transfer matrix method for the detailed derivation of transmissions across the junctions. In Section 3, we demonstrate perfect transmission numerically, showing that the oblique Klein tunneling in NP junctions and the asymmetric Klein tunneling in NPN junctions. Then, we calculate the electrical resistance from the Landauer formula for the NPN junction. Finally, we give a brief conclusion in Section 4.

## 2. Theoretical Formalism

### 2.1. Model

The crystal structure of 8-Pmmn borophene has two sublattices, as illustrated in Figure 1a by different colors. It is made of buckled triangular layers where each unit cell has eight atoms under the symmetry of space group Pmmn (No. 59 in [56]), the so-called 8-Pmmn structure. The tilted Dirac cone emerges from the hexagonal lattice formed by the inner atoms (yellow in Figure 1a) [28]. This hexagonal structure is topologically equivalent to uniaxially strained graphene, and the Hamiltonian of 8-Pmmn borophene around one Dirac point is given by [29,30,36,37].
(1)$$\begin{array}{c} {\hat{H}}_{0}=\upsilon_{x}\sigma_{x}{\hat{p}}_{x}+\upsilon_{y}\sigma_{y}{\hat{p}}_{y}+\upsilon_{t}I_{2\times 2}{\hat{p}}_{y} \end{array}$$
where ${\hat{p}}_{x,y}$ are the momentum operators, $\sigma_{x,y}$ are 2×2 Pauli matrices, and $I_{2\times 2}$ is a 2×2 unit matrix. The anisotropic velocities are $\upsilon_{x}=0.86$$\upsilon_{F},\upsilon_{y}=0.69$$\upsilon_{F},\upsilon_{t}=0.32$$\upsilon_{F},\upsilon_{F}=10^{6}$ m/s [29].The energy dispersion and the corresponding wave functions of ${\hat{H}}_{0}$ are
(2)$$\begin{array}{c} E_{\lambda ,k}=\upsilon_{t}p_{y}+\lambda \upsilon_{x}\sqrt{p_{x}^{2}+\gamma_{1}^{2}p_{y}^{2}},\gamma_{1}=\frac{\upsilon_{y}}{\upsilon_{x}} \end{array}$$
(3)$$\begin{array}{c} \psi_{\lambda ,k}\left(r\right)=\frac{1}{\sqrt{2}}\left[\begin{array}{c} 1\\ \lambda \frac{k_{x}+i\gamma_{1}k_{y}}{\sqrt{k_{x}^{2}+\gamma_{1}^{2}k_{y}^{2}}} \end{array}\right]e^{ik\cdot r} \end{array}$$

Here, λ=±1, denoting the conduction $\left(+1\right)$ and valence $\left(-1\right)$ band, respectively. For 8-Pmmn borophene, the shape of Fermi surface for the fixing energy is elliptical with eccentricity *e* determined by $\upsilon_{x}$, $\upsilon_{y}$ and $\upsilon_{t}$, which differs from the circular shape with radius $E_{F}/\hbar v_{F}$ of graphene. We can rewrite Equation (Equation 2) in following way [39,57]:(4)$$\begin{array}{ccc} & \frac{p_{x}^{2}}{a_{\lambda ,E}}+\frac{{\left(p_{y}+c_{\lambda ,E}\right)}^{2}}{b_{\lambda ,E}}=1 & \end{array}$$(5)$$\begin{array}{ccc} & a_{\lambda ,E}=\frac{\upsilon_{y}^{2}E_{\lambda ,k}^{2}}{\upsilon_{x}^{2}\left(\upsilon_{y}^{2}-\upsilon_{t}^{2}\right)},\;\;\;b_{\lambda ,E}=\frac{\upsilon_{y}^{2}E_{\lambda ,k}^{2}}{{\left(\upsilon_{y}^{2}-\upsilon_{t}^{2}\right)}^{2}},\;\;\;c_{\lambda ,E}=\frac{\upsilon_{t}E_{\lambda ,k}}{\left(\upsilon_{y}^{2}-\upsilon_{t}^{2}\right)} & \end{array}$$

The eccentricity of the Fermi surface can be determined by $e=\sqrt{\upsilon_{x}^{2}-\upsilon_{y}^{2}+\upsilon_{t}^{2}}/\upsilon_{x}$. As a direct consequence, the eccentricity is not depend on the energy and the center of ellipse is at
(6)$$\begin{array}{ccc} \hbar k_{x} & = & 0,\;\;\;\hbar k_{y}=-\frac{\upsilon_{t}E_{\lambda ,k}}{\left(\upsilon_{y}^{2}-\upsilon_{t}^{2}\right)} \end{array}$$

Notice that the center of ellipse is not at the origin and it moves with increasing the Fermi levels. In a NP junction setup, the translation symmetry preserves along the *y* axis, so the $k_{y}$ is always a good quantum number. When the momentum $p_{y}$ is given, the $p_{x}$ in different regions of 8-Pmmn borophene NP junction is
(7)$$\begin{array}{c} p_{x}=\pm \frac{1}{\upsilon_{x}}\sqrt{{\left(E_{\lambda ,k}-\upsilon_{t}p_{y}\right)}^{2}-{\left(\upsilon_{y}p_{y}\right)}^{2}} \end{array}$$

Like the graphene NP junction, one can implement a bipolar NP junction or tunable NPN-type potential barriers in 8-Pmmn borophene by top/back gate voltages, and the potential function of the NP junction (as depicted in Figure 1b) has the form:(8)$$\begin{array}{c} U_{NP}\left(x\right)=\left\{\begin{array}{ccc} V_{0} & , & x>na/2\\ 2V_{0}x/na & , & na/2\leq x\leq na/2\\ -V_{0} & , & x<-na/2 \end{array}\right. \end{array}$$
where $a=\hbar \upsilon_{F}/0.04$ eV is a unit length and n>0∧n∈R. The NPN junction depicted in Figure 1c has the form
(9)$$\begin{array}{c} U_{NPN}\left(x\right)=\left\{\begin{array}{ccccccc} -V_{0} & , & 3na/2+ma & < & x & \\ -2V_{0}\left(x-ma-na\right)/na & , & na/2+ma & \leq & x & \leq & 3na/2+ma\\ V_{0} & , & na/2 & < & x & < & na/2+ma\\ 2V_{0}x/na & , & -na/2 & \leq & x & \leq & na/2\\ -V_{0} & , & & & x & < & -na/2 \end{array}\right. \end{array}$$
where m>0∧m∈R. Next, we will utilize the transfer matrix method to solve the ballistic transport problem in smooth NP/NPN junctions of 8-Pmmn borophene.

### 2.2. Transfer Matrix Method

The transfer matrix method is a powerful tool in the analysis of quantum transport of the massless fermions in 2D Dirac materials [18,58,59]. The central idea lies in that the wave function in one position can be related to those in other positions through a transfer matrix [60].

We adopt a transfer matrix method to study quantum transport in the smooth NP or NPN junction in 8-Pmmn borophene. There are two different matrices in transfer matrix method: one is the transmission matrix and the other is the propagating matrix. Transmission matrix connects the electrons across an interface and the propagating matrix connects the electrons propagating over a distance in the homogeneous regions. As we can see below, the propagating matrix can be derived by the transmission matrix. We define the transmission matrix *T* as follows:(10)$$\begin{array}{ccc} T\left(\begin{array}{c} A_{R_{m+1}}\\ A_{L_{m+1}} \end{array}\right) & = & \left(\begin{array}{c} A_{R_{m}}\\ A_{L_{m}} \end{array}\right) \end{array}$$
where $A_{R_{m}}$ ($A_{L_{m}}$) represents the right (left) traveling wave amplitude in *m* region. The transmission matrix connects the wave function’s amplitude of two different regions. The condition of connecting amplitude coefficients between adjacent regions is the continuity of the wave functions at the interface. We can treat the smooth potential as the sum of infinite slices of junctions and figure out the wave function from the Schrödinger equation. Since the energy dispersion of 8-Pmmn borophene is linear, we only need the continuity condition of the wave functions at the interface. Then, the transmission matrices *T* can be constructed from matrices *M* of each slice,
$$\begin{array}{ccc} & & M\left(k_{m+1},x_{m}\right)\left(\begin{array}{c} A_{R_{m+1}}\\ A_{L_{m+1}} \end{array}\right)=M\left(k_{m},x_{m}\right)\left(\begin{array}{c} A_{R_{m}}\\ A_{L_{m}} \end{array}\right)\\ & & M{\left(k_{m},x_{m}\right)}^{-1}M\left(k_{m+1},x_{m}\right)\left(\begin{array}{c} A_{R_{m+1}}\\ A_{L_{m+1}} \end{array}\right)=\left(\begin{array}{c} A_{R_{m}}\\ A_{L_{m}} \end{array}\right)\\ & & M{\left(k_{m},x_{m}\right)}^{-1}M\left(k_{m+1},x_{m}\right)=T \end{array}$$

Suppose that an *n*-doped region *m* is next to a *p*-doped region m+1 and carriers go through from *n*-doped region to *p*-doped region like in Figure 2a, the wave functions at interface can be connected in the way of
(11)$$\begin{array}{ccc} & & \frac{A_{R_{m+1}}}{\sqrt{2}}\left(\begin{array}{c} 1\\ e^{i\theta_{m+1}} \end{array}\right)e^{ik_{z,m+1}x_{m}+ik_{y}y}+\frac{A_{L_{m+1}}}{\sqrt{2}}\left(\begin{array}{c} 1\\ -e^{-i\theta_{m+1}} \end{array}\right)e^{-ik_{z,m+1}x_{m}+ik_{y}y}\\ & & =\frac{A_{R_{m}}}{\sqrt{2}}\left(\begin{array}{c} 1\\ e^{-i\theta_{m+1}} \end{array}\right)e^{-ik_{z,m}x_{m}+ik_{y}y}+\frac{A_{L_{m}}}{\sqrt{2}}\left(\begin{array}{c} 1\\ -e^{i\theta_{m+1}} \end{array}\right)e^{ik_{z,m}x_{m}+ik_{y}y}. \end{array}$$

Here, we define $k_{x,m}\left(x\right)$ and $\theta_{m}$ as
(12)$$\begin{array}{ccc} k_{x,m}\left(x\right) & = & \frac{1}{\hbar \upsilon_{x}}\sqrt{{\left(-U_{m}\left(x\right)+\hbar \upsilon_{t}k_{y}\right)}^{2}-{\left(\hbar \upsilon_{y}k_{y}\right)}^{2}} \end{array}$$
(13)$$\begin{array}{ccc} e^{i\theta_{m}} & = & \frac{k_{x,m}+i\gamma_{1}k_{y}}{\sqrt{k_{x,m}^{2}+\gamma_{1}^{2}k_{y}^{2}}} \end{array}$$
where $U_{m}\left(x\right)$ is the doping level in *m* region and $k_{x}$ may take positive or negative imaginary values when ${\left(-U_{m}\left(x\right)+\hbar \upsilon_{t}k_{y}\right)}^{2}-{\left(\hbar \upsilon_{y}k_{y}\right)}^{2}<0$. The phase $e^{i\theta_{m}}$ in Equation (Equation 11) is defined as the wave function phase difference between the two sublattices. The sign of the $k_{x}$ defines the propagating direction of the carriers. Without loss of generality, we can take only positive imaginary value for the transmission matrix, which means the positive propagating direction of electrons is defined on right-going state. Here, the potential profile $U_{m}\left(x\right)$ in adjacent regions within NP junction is linear but not rectangular; we treat the potential as a series of step potential to solve the tunneling problems by the transmission matrices. For convenience, we choose $a=\hbar v_{F}/0.04$ eV to be the length unit and 0.01 eV to be the energy unit, where 0.04 eV is the maximum of the doping level.

Then, we rewrite the Equation (Equation 11) to construct the transmission matrices
$$\begin{array}{cc} \left(\begin{array}{cc} e^{-ik_{x,m+1}x_{m}} & e^{ik_{x,m+1}x_{m}}\\ e^{-i\theta_{m+1}}e^{-ik_{x,m+1}x_{m}} & -e^{i\theta_{m+1}}e^{ik_{x,m+1}x_{m}} \end{array}\right)\left(\begin{array}{c} A_{R_{m+1}}\\ A_{L_{m+1}} \end{array}\right) & \\ =\left(\begin{array}{cc} e^{ik_{x,m}x_{m}} & e^{-ik_{x,m}x_{m}}\\ e^{i\theta_{m}}e^{ik_{x,m}x_{m}}\;\;\;\;\;\;\;\;\;\;\;\;\;\;\;\;\;\;\;\;\; & -e^{-i\theta_{m}}e^{-ik_{x,m}x_{m}} \end{array}\right)\left(\begin{array}{c} A_{R_{m}}\;\;\;\;\;\\ A_{L_{m}}\;\;\;\;\; \end{array}\right). \end{array}$$

Therefore, the transmission matrix between *m* and m+1 region is
(14)$$\begin{array}{ccc} \begin{array}{c} T_{m,m+1}^{n\to p} \end{array}\; & \begin{array}{c} = \end{array}\; & \begin{array}{c} {\left(\begin{array}{cc} e^{ik_{x,m}x_{m}} & e^{-ik_{x,m}x_{m}}\\ e^{i\theta_{m}}e^{ik_{x,m}x_{m}} & -e^{-i\theta_{m}}e^{-ik_{x,m}x_{m}} \end{array}\right)}^{-1}\\ \left(\begin{array}{cc} e^{-ik_{x,m+1}x_{m}} & e^{ik_{x,m+1}x_{m}}\\ e^{-i\theta_{m+1}}e^{-ik_{x,m+1}x_{m}} & -e^{i\theta_{m+1}}e^{ik_{x,m+1}x_{m}} \end{array}\right) \end{array} \end{array}$$
while the transmission matrices of the carriers going through from *p*-doped region *m* to *n*-doped region m+1 and between two *n*-doped or *p*-doped region (shown in Figure 2) are
(15)$$\begin{array}{ccc} \begin{array}{c} T_{m,m+1}^{p\to n} \end{array}\; & \begin{array}{c} = \end{array}\; & \begin{array}{c} {\left(\begin{array}{cc} e^{-ik_{x,m}x_{m}} & e^{ik_{x,m}x_{m}}\\ e^{-i\theta_{m}}e^{-ik_{x,m}x_{m}} & -e^{i\theta_{m}}e^{ik_{x,m}x_{m}} \end{array}\right)}^{-1}\\ \left(\begin{array}{cc} e^{ik_{x,m+1}x_{m}} & e^{-ik_{x,m+1}x_{m}}\\ e^{i\theta_{m+1}}e^{ik_{x,m+1}x_{m}} & -e^{-i\theta_{m+1}}e^{-ik_{x,m+1}x_{m}} \end{array}\right) \end{array} \end{array}$$
(16)$$\begin{array}{ccc} \begin{array}{c} T_{m,m+1}^{n\to n} \end{array}\; & \begin{array}{c} = \end{array}\; & \begin{array}{c} {\left(\begin{array}{cc} e^{ik_{x,m}x_{m}} & e^{-ik_{x,m}x_{m}}\\ e^{i\theta_{m}}e^{ik_{x,m}x_{m}} & -e^{-i\theta_{m}}e^{-ik_{x,m}x_{m}} \end{array}\right)}^{-1}\\ \left(\begin{array}{cc} e^{ik_{x,m+1}x_{m}} & e^{-ik_{x,m+1}x_{m}}\\ e^{i\theta_{m+1}}e^{ik_{x,m+1}x_{m}} & -e^{-i\theta_{m+1}}e^{-ik_{x,m+1}x_{m}} \end{array}\right) \end{array} \end{array}$$
(17)$$\begin{array}{ccc} \begin{array}{c} T_{m,m+1}^{p\to p} \end{array}\; & \begin{array}{c} = \end{array}\; & \begin{array}{c} {\left(\begin{array}{cc} e^{-ik_{x,m}x_{m}} & e^{ik_{x,m}x_{m}}\\ e^{-i\theta_{m}}e^{-ik_{x,m}x_{m}} & -e^{i\theta_{m}}e^{ik_{x,m}x_{m}} \end{array}\right)}^{-1}\\ \left(\begin{array}{cc} e^{-ik_{x,m+1}x_{m}} & e^{ik_{x,m+1}x_{m}}\\ e^{-i\theta_{m+1}}e^{-ik_{x,m+1}x_{m}} & -e^{i\theta_{m+1}}e^{ik_{x,m+1}x_{m}} \end{array}\right) \end{array} \end{array}$$

For the case of NPN junction, a trapezoidal potential profile as in Figure 1c, we can also treat the trapezoidal potential into infinite slices of connected step potentials. The transmission matrices define at the interface between each step potentials. Multiplying all the transmission matrices would give the propagation matrices,
(18)$$\begin{array}{ccc} & T_{all}=T_{0,1}^{n\to n}T_{1,2}^{n\to n}\ldots T_{k-1,k}^{n\to n}T_{k,k+1}^{n\to p}T_{k+1,k+2}^{p\to p}\ldots \times & \\ & T_{k^{\prime }-1,k^{\prime }}^{p\to p}T_{k^{\prime },k^{\prime }+1}^{p\to p}\ldots T_{k^{\prime \prime }-1,k}^{p\to p}T_{k^{\prime \prime },k^{\prime \prime }+1}^{p\to n}T_{k^{\prime \prime }+1,k^{\prime \prime }+2}^{n\to n}\ldots T_{m-2,m-1}^{n\to n}T_{m-1,m}^{n\to n} \end{array}$$

Then, we reach the formula
(19)$$\begin{array}{c} T_{all}\left(\begin{array}{c} A_{R_{m}}\\ A_{L_{m}} \end{array}\right)=\left(\begin{array}{c} A_{R_{0}}\\ A_{L_{0}} \end{array}\right) \end{array}$$

When incident electrons go from the leftmost side of the NPN junction to the rightmost side, there are no reflection states in the rightmost side, i.e., $A_{L_{m}}=0$. We can connect the amplitude of incident states to the amplitude of reflection states
$$\begin{array}{ccc} \left(\begin{array}{cc} T_{11} & T_{12}\\ T_{21} & T_{22} \end{array}\right)\left(\begin{array}{c} A_{R_{m}}\\ 0 \end{array}\right) & = & \left(\begin{array}{c} A_{R_{0}}\\ A_{L_{0}} \end{array}\right)\\ \frac{A_{R_{m}}}{A_{R_{0}}} & = & \frac{1}{T_{11}} \end{array}$$

Finally, the transmission probability is $T={\left|t\right|}^{2}={\left|A_{R_{m}}/A_{R_{0}}\right|}^{2}={\left|1/T_{11}\right|}^{2}$.

There is a trick in constructing the propagation matrices from the transmission matrices. As shown in Figure 3, the incident states at the left-hand side of the junction have a different Fermi surface from the transmitted states at the right-hand side in the NP junction. Suppose the NP junction is sharp. The good quantum number $k_{y}$ should be restricted between the top dotted green line and the middle dotted green line, since the incident states and the transmitted states are propagating only in this scenario. While supposing the NP junction is smooth, the Fermi surface in the region of varying potential would shrink to the Dirac point, and the $E_{\lambda ,k}$, $a_{\lambda ,E}$, $b_{\lambda ,E}$, and $c_{\lambda ,E}$ from Equation (5) reduce to zero as well. Therefore, $k_{x}$ vanishes to diverge the transmission matrices when the carriers approaching the NP junction center. However, we could play a trick by properly segmenting the region of varying potential and jumping the diverging point. The trick lies in the fact that the carriers would not experience any singularity when going through an infinitesimal interval around the diverging point. For instance, the transmission matrix at the Dirac point cannot be well defined with incident states $k_{y}=0$, whereas the carriers are well-defined decay states at the Dirac point. We can ignore the decay states of the carriers going through infinitesimal intervals around the Dirac point, and it would eliminate any possible ambiguity.

## 3. Results and Discussions

In this section, we present the numerical results for the transmission probability and electrical conduction of the massless Dirac fermions across the borophene NP junction and NPN junction.

### 3.1. The Oblique Klein Tunneling in Smooth NP Junctions

Various smooth NP junctions with fixing *n*/*p* doping level but different slopes are depicted in Figure 4a. We set the length of the varying region in different NP junctions as 6.25
*a*, 12.5
*a*, 25 *a*, and 50 *a*, respectively, where $a=\hbar \upsilon_{F}/0.04$ eV, and plot the angular transmission probability for different NP junctions. As shown in Figure 4b, the shaper the NP junction is, the wider the angular transmission probability spans. This phenomenon is caused by the decay states in the varying region and is similar to the graphene smooth NP junction. In the varying region, ${\left(-U_{m}\left(x\right)+\hbar \upsilon_{t}k_{y}\right)}^{2}-{\left(\hbar \upsilon_{y}k_{y}\right)}^{2}<0$, so that the propagating states degenerate to the decaying states when the carriers gradually approach the junction’s center. Therefore, the transmission probability increases with increasing the slope of potential in the varying region. If we take $k_{y}=0$, i.e., the normal incident case, we can see the perfect transmission, the Klein tunneling.

Figure 5 shows that the angular transmission amplitude of the *k* vector is different from the one of group velocity. The actual incident angle across the junction is based on the group velocity of carriers. The actual angular transmission probability for group velocity shown in Figure 5 indicates a rotation of the Klein tunneling, the oblique Klein tunneling. It means that the perfect transmission does not occur in the normal incident but with a nonzero angle $\theta_{K}$.

The value of $\theta_{K}$ can be determined from the elliptical Fermi surface of 8-Pmmn borophene. The angle for the group velocity is $\theta_{v}=\arctan\left[v_{y}\left(\varepsilon ,k_{y}\right)/v_{x}\left(\varepsilon ,k_{y}\right)\right]$, where $v_{y}\left(\varepsilon ,k_{y}\right)$ and $v_{x}\left(\varepsilon ,k_{y}\right)$ can be obtained by
(20)$$\begin{array}{ccc} v_{x}\left(\varepsilon ,k_{y}\right) & = & \frac{\partial E_{\lambda ,k}}{\hbar \partial k_{x}}=\frac{\lambda k_{x}\upsilon_{x}}{\sqrt{k_{x}^{2}+\gamma_{1}^{2}k_{y}^{2}}} \end{array}$$
(21)$$\begin{array}{ccc} v_{y}\left(\varepsilon ,k_{y}\right) & = & \frac{\partial E_{\lambda ,k}}{\hbar \partial k_{y}}=\upsilon_{t}+\frac{\lambda \gamma_{1}^{2}k_{y}\upsilon_{x}}{\sqrt{k_{x}^{2}+\gamma_{1}^{2}k_{y}^{2}}} \end{array}$$

Combined with above equations and let $k_{y}=0$, we can find the angle of Klein tunneling for group velocity,
(22)$$\begin{array}{c} \theta_{K}=\arctan\left(\frac{\upsilon_{t}}{\upsilon_{x}}\right)\approx 20.4^{\circ } \end{array}$$

This oblique Klein tunneling can also be found in sharp NP junctions of 8-Pmmn borophene [39,53].

### 3.2. The Asymmetric Klein Tunneling in the Smooth NPN Junctions

The NPN junction, as shown in the Figure 1c, can be seen as a trapezoid potential barrier. We set the length of the varying regions as 6.25
*a* and the length of the flat potential barrier as 12.5
*a*.

In Figure 6, we plot the transmission probability depending on different doping levels and $k_{y}$. Note that *n*- and *p*-regions have the same absolute value of doping level. We can see the Klein tunneling in several branches. The number of branches increases by lifting the doping level, which could also be observed in the graphene NPN junctions [16,18]. We can see the Klein tunneling is asymmetric. The asymmetric Klein tunneling results from the carriers’ chirality and anisotropy [61]. It is not surprising to see it here because the carriers of 8-Pmmn have both chirality and anisotropy.

The blue lines in Figure 6a denote the forbidden zones, where the transmission probability vanishes. The equation of the boundary of the forbidden zone is $k_{y}=\pm \varepsilon_{doping}/\hbar \left(\upsilon_{t}+\upsilon_{y}\right)$. There are two types of the forbidden zone: (I) the no-incident zone and (II) the vanishing transmitted zone. In the no-incident zone $k_{y}\geq \varepsilon_{doping}/\hbar \left(\upsilon_{t}+\upsilon_{y}\right)$, there is no incident states since the parameters $k_{y}$ and doping level is beyond the Dirac cone; in the vanishing transmitted zone $k_{y}\leq -\varepsilon_{doping}/\hbar \left(\upsilon_{t}+\upsilon_{y}\right)$, the transmitted carriers severely decay in the region of barrier.

Next, we fix the bottom edge and the height of the trapezoid potential (NPN junction) and plot the transmission probability versus the potential’s top edge. When the top edge’s length varies from 0 to the bottom edge’s length, the NPN junction experiences a change from triangle potential to trapezoid potential and finally to a square potential. We can see from Figure 7, the number of branches increases with increasing the top edge’s length. It is somehow counterintuitive that the square potential favors Klein tunneling more than the triangle potential. The reason is that the carriers would have more chances to degenerate to decaying states when incident into a slope of potential, in fact, a smooth NP junction.

### 3.3. The Electrical Resistance of the Smooth NPN Junctions

One can create the NPN junction by implementing a design with two electrostatic gates, a global back gate and a local top gate. A back voltage applied to the back gate could tune the carrier density in the borophene sheet, whereas a top voltage applied to the top gate could tune the density only in the narrow strip below the gate. These two gates can be controlled independently [62].

To clarify the effect of the Klein tunneling on the transport property, here, we discuss the electrical conduction of the NPN junction in 8-Pmmn borophene. In the ballistic regime, we apply the Landauer–Buttiker formula $G=2e^{2}MT/h$ to calculate the electrical conductance [63]. In our setup, the Landauer formula can be written as [64]
(23)$$\begin{array}{c} G_{fet}=\frac{4e^{2}}{h}\sum_{ch.}T_{ch}\approx \frac{4e^{2}}{h}\int_{k_{y\min}}^{k_{y\max}}\frac{dk_{y}}{2\pi /W}T\left(k_{y}\right) \end{array}$$
where $k_{y\max}=\varepsilon_{doping}/\hbar \left(\upsilon_{t}+\upsilon_{y}\right)$ and $k_{y\min}=\varepsilon_{doping}/\hbar \left(\upsilon_{t}-\upsilon_{y}\right)$.

We choose the width of the junction W=10μm and calculate the electrical resistance by the Landauer formula. To reveal the link of the resistance with the Klein tunneling, we plot the transmission probability versus the doping level in Figure 8a and the resistance depending on doping levels in Figure 8b. We can see the resistance oscillation when increasing the doping level from 0 to 0.08 eV. The oscillation pattern indicates the effect of the Klein tunneling. When the doping level varies from 0 to −0.08 eV, the NPN junction becomes a NNN junction so that the curves of resistance are flat in the negative doping regime.

## 4. Conclusions

This work investigates the transport properties of massless fermions in the smooth 8-Pmmn borophene NP and NPN junctions by the transfer matrix method. Compare with the sharp junction, the smooth NP junction also shows that the oblique Klein tunneling induced by the tilted Dirac cones. We can calculate from the parameters of the Hamiltonian that the angle of oblique Klein tunneling is $20.4^{\circ }$. We also show the branches of the NPN tunneling in the phase diagram, which indicates the asymmetric Klein tunneling. The physical origin of the asymmetric Klein tunneling lies in the chirality and anisotropy of the carriers, and we can verify the asymmetric Klein tunneling experimentally by analyzing the pattern of the electrical resistance oscillation. For the oblique Klein tunneling, we have discussed the experimental feasibility in detail in our previous study [39]. The present numerical demonstration in smooth junctions proves the effectiveness of our previous discussion and favors the observation in future experiments. 

## Figures and Tables

**Figure 1 nanomaterials-11-01462-f001:**
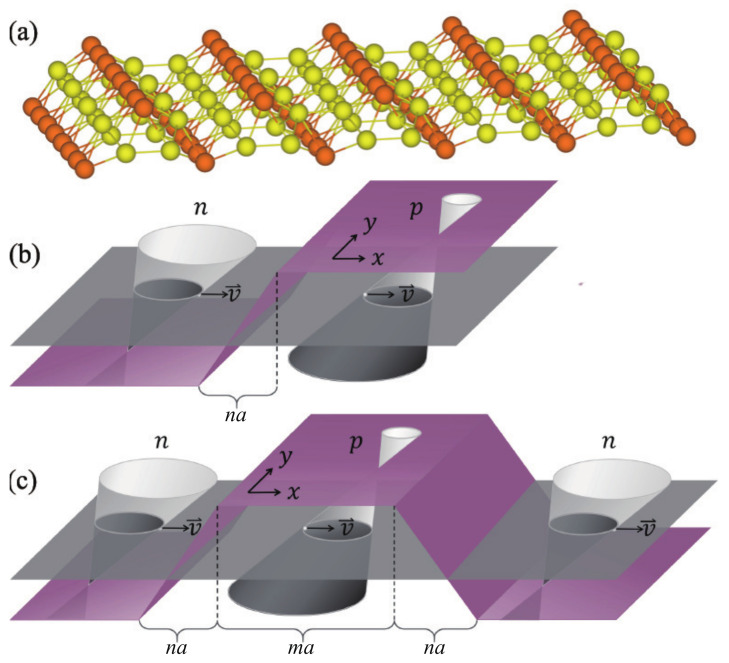
(**a**) Crystal structure of 8-Pmmn borophene. The unit cell of 8-Pmmn borophene contains two types of nonequivalent boron atoms, the ridge atoms (orange) and the inner atoms (yellow). (**b**) The schematic diagram of the smooth NP junction in 8-Pmmn borophene. Note that the true tilted Dirac cone is along *y* direction but *x* direction. (**c**)The schematic diagram of the smooth NPN junction in 8-Pmmn borophene. Here, we choose n=6.25 and m=12.5 for the numerical calculations.

**Figure 2 nanomaterials-11-01462-f002:**
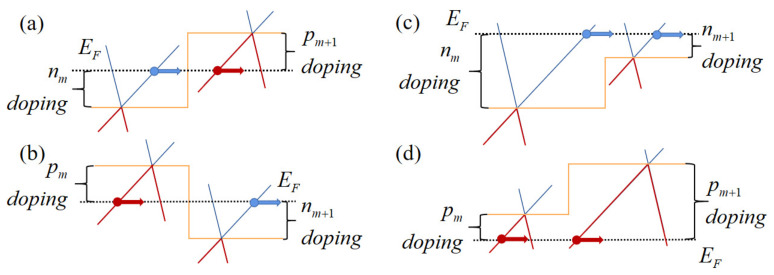
Potential profile of (**a**) NP junction, (**b**) PN junction, (**c**) NN junction, and (**d**) PP junction in each slice of the junctions.

**Figure 3 nanomaterials-11-01462-f003:**
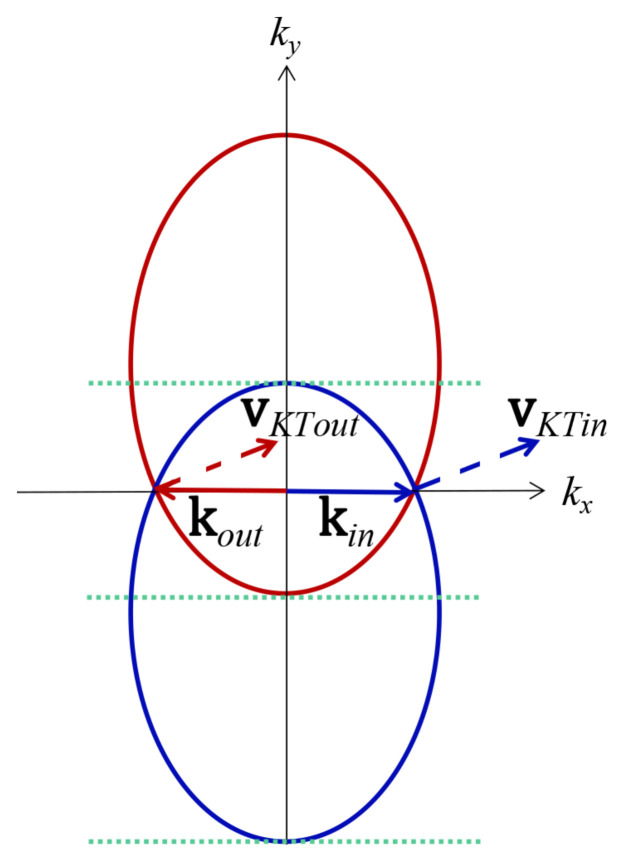
Fermi surface at different doped regions $\pm \varepsilon_{doping}$. The blue (red) ellipse represents the electron (hole) Fermi surface in *n*-doped (*p*-doped) region. The solid vectors $k_{in}$ (blue) and $k_{out}$ (red) are the wave vector of incident carriers and transmitted carriers, respectively. The dashed vectors $v_{KTin}$ (blue) and $v_{KTout}$ (red) are the group velocity of incident carriers and transmitted carriers, respectively. The green dotted lines indicate the values of the good quantum number $k_{y}$ posed restrictions for the NP junction and the NPN junction.

**Figure 4 nanomaterials-11-01462-f004:**
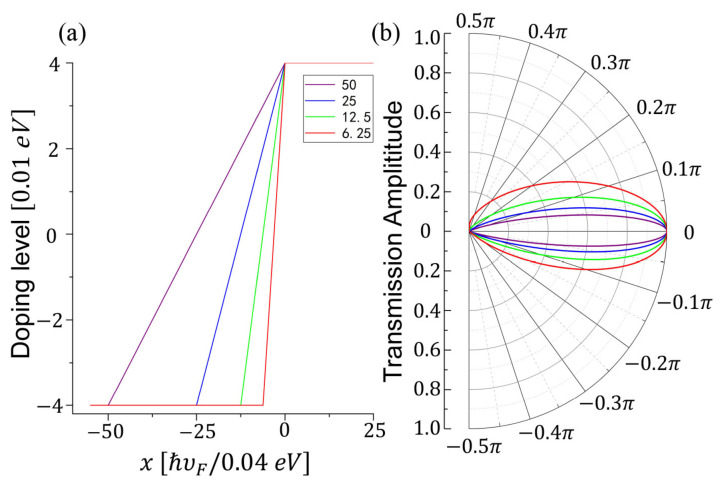
(**a**) Potential profile of smooth NP junctions and (**b**) the angular behavior of the transmission probability for different NP junctions corresponding to different colors at (**a**).

**Figure 5 nanomaterials-11-01462-f005:**
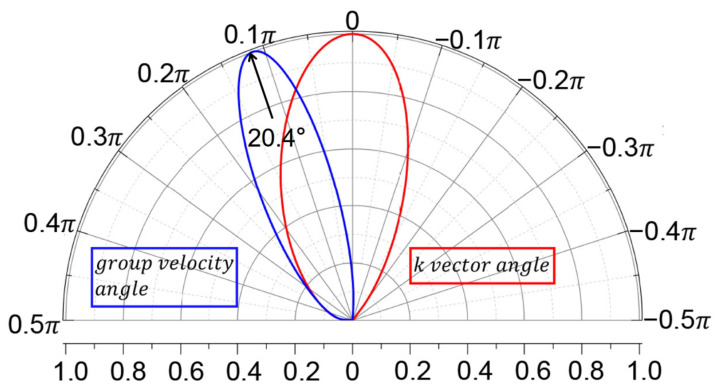
Angular transmission amplitude for *k* vector (red) and for group velocity (blue). The doping level is 0.04 eV and the length of varying region is 6.25
*a*.

**Figure 6 nanomaterials-11-01462-f006:**
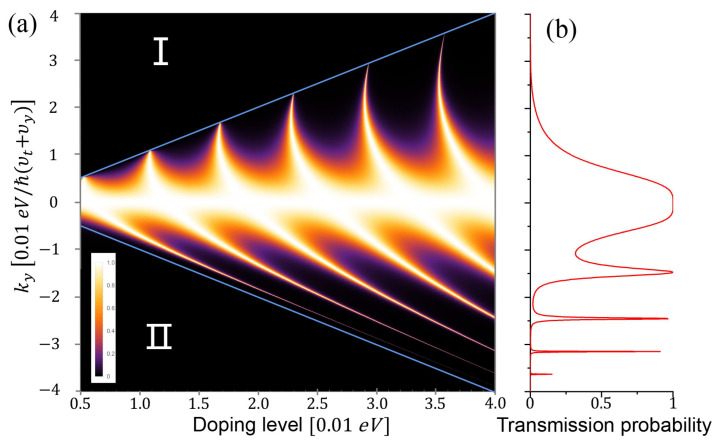
(**a**) Transmission probability versus the doping level and the $k_{y}$ in NPN junction. Blue lines denote the forbidden zone, where transmission probability vanishes, and there are only the decaying states in the *p*-doped region. (**b**) The transmission probability depending on $k_{y}$ when the doping level is 4×0.01 eV.

**Figure 7 nanomaterials-11-01462-f007:**
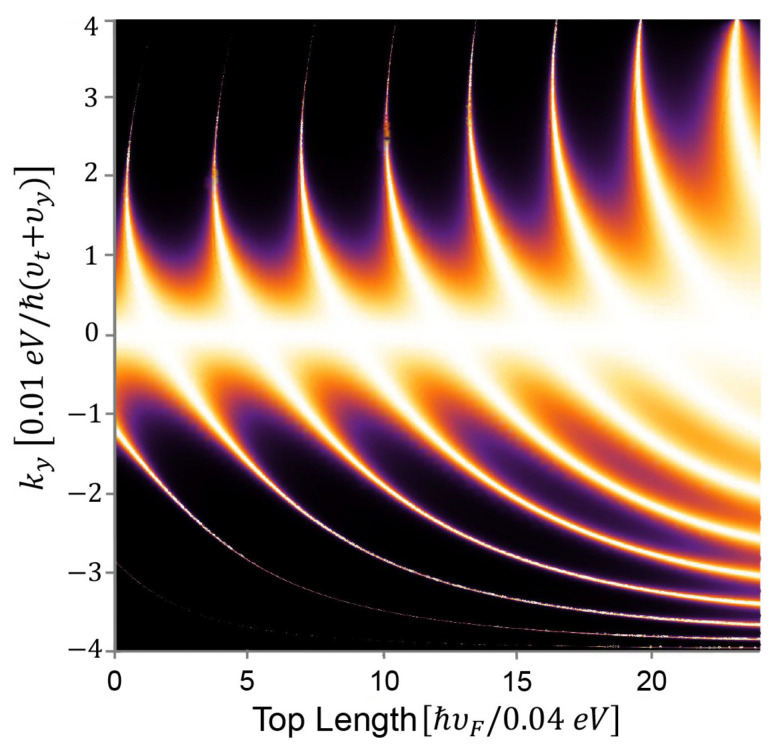
Transmission probability depends on top edge of the trapezoid potentials. The top edge varies from 0 to 24 *a* and the bottom edge is fixed as 25 *a*. The height of the trapezoid potentials or the absolute value of n/p doping level is fixed as 0.04 eV.

**Figure 8 nanomaterials-11-01462-f008:**
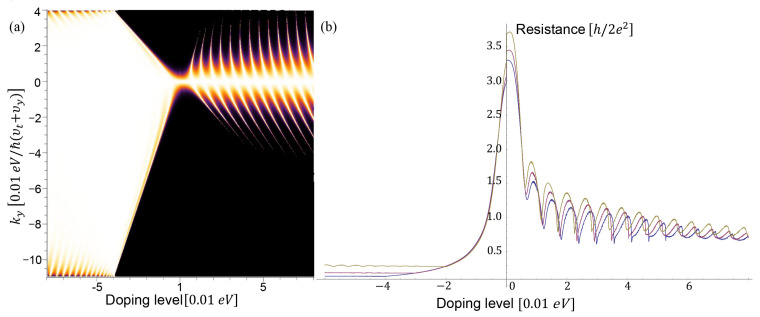
(**a**) Transmission probability depends on $k_{y}$ and the height of the trapezoid potentials (doping levels of the NPN junctions). The top edge’s length is 12.5
*a* and the bottom edge’s length is 25 *a*. The doping level of *n*-doped region (outside the NPN junction) is set −0.04 eV. (**b**) The electrical resistance of the NPN junction depending on the doping level.

## Data Availability

The study did not report any data.

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
