# Peer review of "Oblique and Asymmetric Klein Tunneling across Smooth NP Junctions or NPN Junctions in 8-Pmmn Borophene"

_nanomaterials, 2021, doi:10.3390/nano11061462_

Round 1
Reviewer 1 Report
The paper contains a significant theoretical advance in the understanding of the electronic structure of smooth NP or NPN junctions in 8-Pmmn borophene. In my opinion, the work can be accepted in its present form. However, it would be appropriate to discuss in more detail the possibility of applying the proposed method to other similar materials.
Author Response
Point 1: The paper contains a significant theoretical advance in the understanding of the electronic structure of smooth NP or NPN junctions in 8-Pmmn borophene. In my opinion, the work can be accepted in its present form. However, it would be appropriate to discuss in more detail the possibility of applying the proposed method to other similar materials.
Response 1: We appreciate the referee's recommendation! We will apply the proposed method to other materials in our next work.
Reviewer 2 Report
The manuscript presents a solid theoretical study. The authors studied the transmission properties of anisotropic and tilted massless Dirac fermions across NP and NPN junctions in 8-Pmmn borophene. In my opinion, the manuscript does not contain serious drawbacks. It is well written and provides interesting results. The chosen theoretical model is adequate. The authors have detailed everything so that interested parties can repeat the presented procedures. The results obtained may be of interest for further experimental studies. Therefore, I think the manuscript can be accepted for publication in Nanomaterials.
Author Response
Point 1: Therefore, I think the manuscript can be accepted for publication in Nanomaterials.
Response 1: We appreciate the referee's recommendation!
Reviewer 3 Report
The manuscript titled: "Oblique and asymmetric Klein tunneling across smooth NP junctions or NPN junctions in 8-Pmmn borophene" presenting a standard numerical investigation on the Klein tunneling of massless fermions across the smooth NP junctions and NPN junctions realized for the 8-Pmmn borophene. It is found that the oblique Klein tunneling retains in borophene due to the tilted Dirac cone. Based on the Hamiltonian parameters of 8-Pmmn borophene the research predicts the angle of oblique Klein tunneling is predicted to be 20.4. It is also found that asymmetric Klein tunneling is induced by the chirality and anisotropy of the carriers.
Despite the scientific problem discussed in the manuscript is related to nanoscience and nanomaterials and may be valuable for the scientific community, there are several critical disadvantages in this work. The main part of the work consists of the mathematical calculations which are not directly in the scope of the Nanomaterials journal. More specific journal can be considered.
The novelty of the work is not clear as well. For instance, the Authors claim that they "developed the transfer matrix method" at the same time they use to say that "a transfer matrix method is a powerful tool in the analysis of quantum transport of the massless fermions in 2D Dirac materials". Hence, this work implements this approach for the borophene case only. Moreover, the Authors have already considered Oblique Klein tunneling in 8−Pmmn borophene p−n junctions [Phys. Rev. B 97, 235440, 2018].
Besides, there are few additional comments which can be found below.
Comment 1. The manuscript should be carefully proofread. Particularly, the writing should be improved. There are also many typos. References are not adjusted to a uniform format. The text in the abstract, the last section of the introduction part, and the conclusion part is identical (simply copy-pasted).
Comment 2. The introduction part too short and not informative. In addition, a significant part of the introduction is devoted to the description of the results, which is inappropriate in the introduction.
Comment 3. “The analysis of the pattern of the oscillation of electrical resistance would help verify the existence of asymmetric Klein tunneling experimentally.”
The work will be more solid if experimental verification of the proposed method will be provided, or the methodology will be described more detailed.
Author Response
Point 1: The main part of the work consists of the mathematical calculations which are not directly in the scope of the Nanomaterials journal. More specific journal can be considered.
Response 1: The research of nanomaterials is a diverse field, with 2D materials as the typical example. Our manuscript focuses on the tunneling properties of the junctions based on 2D materials, which is helpful to the understanding and applications of 2D materials. We believe our work will contribute to the research of nanoscience and is suitable for Nanomaterials.
Point 2: For instance, the Authors claim that they "developed the transfer matrix method" at the same time they use to say that "a transfer matrix method is a powerful tool in the analysis of quantum transport of the massless fermions in 2D Dirac materials". Hence, this work implements this approach for the borophene case only.
Response 2: We thank the constructive advice from the referee. To be rigorous, we choose the word “adopted” instead of “developed".
Point 3: Moreover, the Authors have already considered Oblique Klein tunneling in 8−Pmmn borophene p−n junctions [Phys. Rev. B 97, 235440, 2018].
Response 3: The present manuscript is different from our previous paper for two main reasons: (i) The SHARP junctions are ideal cases, where SHARP interfaces pose a tough challenge to fabrication. Therefore, the SMOOTH p-n junctions are more practical structures. (ii) The method used in SHARP junctions is not applicable for SMOOTH junctions. We need to adopt a different method, the transfer matrix, to calculate the Klein tunneling.
Point 4: The manuscript should be carefully proofread. Particularly, the writing should be improved. There are also many typos. References are not adjusted to a uniform format.
Response 4: We appreciate the referee’s suggestion! We carefully proofread our manuscript again and format the references in a consistent style.
Point 5: The introduction part too short and not informative. In addition, a significant part of the introduction is devoted to the description of the results, which is inappropriate in the introduction…. The text in the abstract, the last section of the introduction part, and the conclusion part is identical (simply copy-pasted).
Response 5: We appreciate the referee’s suggestion! Considering the research of borophene is a new and rising field, we have already included the necessary information about the background and motivation within a compact style. Following the suggestion of the referee, we rewrite the conclusion part.
Point 6: “The analysis of the pattern of the oscillation of electrical resistance would help verify the existence of asymmetric Klein tunneling experimentally.”
The work will be more solid if experimental verification of the proposed method will be provided, or the methodology will be described more detailed.
Response 6: We agree that it would be better to analyze the pattern of the oscillation. And it is an interesting topic to perform this analysis in the absence or presence of a magnetic field, which leaves for future work. On the experimental side, we expect future verification as done in graphene.
Reviewer 4 Report
The authors provide a theoretical modeling of transmission through regions of sharply changing doping level in a new 2D material, borophene. The presentation is excellent, as are the figures showing the results. Minor changes are requested.
When introduced as one of a family of borophene crystal structures, the description “8-Pmmn” may not be familiar to non-specialists. Please give some background and describe the index sequence in this designation.
Many equations after (9) do not have an equation number, and in case someone wants to refer to them this will be difficult. Please enumerate all equations or sets of equations. The phase factor in (10) may be quite common but is not defined. It would help to describe its significance.
Fig. 4(b) needs a caption.
Line 144: ”The trick lies in the fact that the carriers would not experience any singularity when going through an infinitesimal interval around the diverging point.” It would be instructive to explain why this is considered a special occurrence. I like describing this as a “trick” but it is not clear why.
Author Response
Point 1: When introduced as one of a family of borophene crystal structures, the description “8-Pmmn” may not be familiar to non-specialists. Please give some background and describe the index sequence in this designation.
Response 1: We appreciate the referee’s suggestion and add an explanation of the description “8-Pmmn” in the introduction.
Point 2: Many equations after (9) do not have an equation number, and in case someone wants to refer to them this will be difficult. Please enumerate all equations or sets of equations. The phase factor in (10) may be quite common but is not defined. It would help to describe its significance. …Fig. 4(b) needs a caption.
Response 2: We appreciate the referee’s suggestion! We enumerate the equations after (9) and add the definition of the phase factor. The caption of Fig. 4(b) has been rewritten.
Point 3: Line 144: “The trick lies in the fact that the carriers would not experience any singularity when going through an infinitesimal interval around the diverging point.” It would be instructive to explain why this is considered a special occurrence. I like describing this as a “trick” but it is not clear why.
Response 3: We add an instance for explaining the trick as follows:
"For instance, the transmission matrix at the Dirac point cannot be well defined with incident states $k_{y}=0$, whereas the carriers are well-defined decay states at the Dirac point. So we can ignore the decay states of the carriers going through infinitesimal intervals around the Dirac point, and it would eliminate any possible ambiguity."
Round 2
Reviewer 3 Report
The authors replied to several comments on the choice of the journal and on the difference of their work from the previous one. However, the rest comments were not clearly addressed. Considering high standards of the Nanomaterials journal, I cannot suggest the publication of the manuscript in its current form.
The introduction part has not been improved and the major part still consists of the authors findings rather than a literature review. The text in the abstract, the last section of the introduction part, and the conclusion part is still almost identical.
The introduction still consists of outdated references. There are many new studies on 8-Pmmn borophene and p-n junctions based on it. On the first page of google, the following can be found: Physics Letters A, 384, 2020, 126612; Nanotechnology, 32, 025205, 2021; Phys. Rev. B, 99, 2019, 155418.
The authors mentioned that the experimental verification of similar work has been done for graphene and they will soon conduct experiments for borophene. However, there is still no discussion on the methodology for possible experimental verification of their results. In this case, the work is of interest only to a specific community of researchers and may not be in line with the scope of the Nanomaterials journal.
Author Response
Point 1: The authors replied to several comments on the choice of the journal and on the difference of their work from the previous one. However, the rest comments were not clearly addressed. Considering high standards of the Nanomaterials journal, I cannot suggest the publication of the manuscript in its current form.
Response 1: We appreciate the referee for reviewing our manuscript by adopting the high standards. Below we further improve our manuscript according to the referee's comments and hope the revised manuscript would satisfy the referee.
Point 2: The introduction part has not been improved and the major part still consists of the authors findings rather than a literature review. The text in the abstract, the last section of the introduction part, and the conclusion part is still almost identical.
Response 2: We appreciate the referee's comments on the writing style of the manuscript. We originally pursued a concise and compact style by considering the fast development of the borophene field with references, i.e., from the concept of Klein tunneling directly to the relevant studies of Klein tunneling in 8-Pmmn borophene, and the problem to address. As pointed out by the referee, we did not give a literature review on the 8-Pmmn borophene, which may be important to the wide audience. Following the referee's suggestion, we add a literature review on the various properties of 8-Pmmn borophene in the second paragraph of the introduction [see list of changes (1)], i.e., "These unique Dirac fermions attracts people to explore the various physical properties such as strain-induced pseudomagnetic field [PhysRevB.94.165403 (2016)], anisotropic density-density response [PhysRevB.96.035410,PhysRevB.98.195415,PhysRevB.98.235430,PhysRevLett.125.116802], optical conductivity [PhysRevB.96.155418, PhysRevB.103.165415], modified Weiss oscillation [PhysRevB.96.235405], borophane and its tight-binding model [PhysRevB.97.125424], nonlinear optical polarization rotation [PhysRevB.97.205420], oblique Klein tunneling [PhysRevB.97.235440,PhysRevB.100.195139, j.physleta.2020.126612], few-layer borophene [PhysRevB.98.054104,PhysRevB.98.115413, ], intense light response [PhysRevB.99.035415,PhysRevB.100.125302], RKKY interaction [PhysRevB.99.155418,/j.jmmm.2019.165631], anomalous caustics [Zhang_2019_New_J._Phys._21_103052], electron-phonon coupling [PhysRevB.100.024503], valley-contrast behaviors [PhysRevB.102.045417, 10.1088/1361-6528/abbbd7], Andreev reflection [PhysRevB.102.045132], and so on. " In addition, to avoid sentence repetition, we do our best to reorganize the conclusion part.
Point 3: The introduction still consists of outdated references. There are many new studies on 8-Pmmn borophene and p-n junctions based on it. On the first page of google, the following can be found: Physics Letters A, 384, 2020, 126612; Nanotechnology, 32, 025205, 2021; Phys. Rev. B, 99, 2019, 155418.
Response 3: We appreciate the referee for providing the relevant references which are cited in the literature review of borophene in the revised manuscript [see list of changes (1)]. But we are sorry to disagree with the wording "outdated references," and please allow us to explain this point: To us, the original references contributing to science is the most crucial inheritance for the research, which should not be overlooked.
Point 4: The authors mentioned that the experimental verification of similar work has been done for graphene and they will soon conduct experiments for borophene. However, there is still no discussion on the methodology for possible experimental verification of their results. In this case, the work is of interest only to a specific community of researchers and may not be in line with the scope of the Nanomaterials journal.
Response 4: As theoretical researchers, we cannot conduct experiments for borophene. On the experimental feasibility, we have performed the detailed discussions in the previous paper [PhysRevB.97.235440 (2018)] as mentioned by the referee in the last comment. We believe the successful transport measurement of Klein tunneling in graphene [PhysRevB.86.155412 (2012), Nano Lett. 12, 4460 (2012), Appl. Phys. Lett. 106, 013112 (2015), Science 353, 1522 (2016)] can also be applied to 8-Pmmn borophene. Following the analytical study of the sharp junction [PhysRevB.97.235440 (2018)], this work further demonstrates the oblique Klein tunneling numerically in the smooth junctions, which should be more practical to the experimental scientists for the observation. In the revised manuscript, we add one sentence to briefly discuss the experimental verification of our results in the Conclusion section [see list of changes (3)], i.e., "For the oblique Klein tunneling, we have discussed the experimental feasibility in detail in our previous study[PhysRevB.97.235440 (2018)]. The present numerical demonstration in smooth junctions proves the effectiveness of our previous discussion and favors the observation in future experiments."
List of changes:
The suggestions have been incorporated into our revised manuscript:
(1) We add “This unique Dirac fermions attracts people to explore the various physical properties such as strain-induced pseudomagnetic field \cite{ref-21}, anisotropic density-density response \cite{ref-29,ref-71,ref-72,ref-73}, optical conductivity \cite{ref-74,ref-75}, modified Weiss oscillation \cite{ref-28,ref-41}, borophane and its tight-binding model\cite{ref-41}, nonlinear optical polarization rotation \cite{ref-76}, oblique Klein tunneling\cite{ref-19,ref-77,ref-78}, few-layer borophene\cite{ref-79,ref-80} , intense light response \cite{ref-81,ref-82}, RKKY interaction \cite{ref-83,ref-84}, anomalous caustics \cite{ref-85}, electron-phonon coupling \cite{ref-86}, valley-contrast behaviors \cite{ref-87,ref-88}, Andreev reflection \cite{ref-89}, and so on.”at line 37.
(2) We replaced “Some recent researches have reported the oblique Klein tunneling, the deviation of the perfect transmission direction to the normal direction of the interface, induced by the anisotropic massless Dirac fermions or the tilted Dirac cone\cite{ref-18,ref-19}”with “The oblique Klein tunneling, the deviation of the perfect transmission direction to the normal direction of the interface, is induced by the anisotropic massless Dirac fermions or the tilted Dirac cone\cite{ref-18,ref-19}.” at line 43.
(3) We replaced the conclusion “This study adopted the transfer matrix method to investigate the transport properties of massless fermions in the smooth NP junctions and NPN junctions of 8-Pmmn borophene. Similar to the sharp junction, the smooth NP junction also shows that the oblique Klein tunneling induced by the tilted Dirac cones. The angle of oblique Klein tunneling, $20.4^\circ$ , is a constant determined by the Hamiltonian parameters of 8-Pmmn borophene. There are branches of the tunneling in the NPN junction in the phase diagram, which indicates the asymmetric Klein tunneling. The asymmetric Klein tunneling is induced by the chirality and anisotropy of the carriers, resulting in an oscillation of the electrical resistance. One may analyze the pattern of the oscillation of electrical resistance and verify the existence of asymmetric Klein tunneling experimentally.” with “This work investigates the transport properties of massless fermions in the smooth 8-Pmmn borophene NP and NPN junctions by the transfer matrix method. Compare with the sharp junction, the smooth NP junction also shows that the oblique Klein tunneling induced by the tilted Dirac cones. We can calculate from the parameters of the Hamiltonian that the angle of oblique Klein tunneling is $20.4^\circ$. We also show the branches of the NPN tunneling in the phase diagram, which indicates the asymmetric Klein tunneling. The physical origin of the asymmetric Klein tunneling lies in the chirality and anisotropy of the carriers, and we can verify the asymmetric Klein tunneling experimentally by analyzing the pattern of the electrical resistance oscillation. For the oblique Klein tunneling, we have discussed the experimental feasibility in detail in our previous study\cite{ref-19}. The present numerical demonstration in smooth junctions proves the effectiveness of our previous discussion and favors the observation in future experiments.” at line 239.
(4) We add references
“31. Z. Jalali-Mola and S. A. Jafari. Tilt-induced kink in the plasmon dispersion of two-dimensional Dirac electrons. Phys. Rev. B. 2018, 98, 195415.”
“32. Z. Jalali-Mola and S. A. Jafari. Kinked plasmon dispersion in borophene-borophene and borophene-graphene double layers. Phys. Rev. B. 2018, 98, 235430.”
“33. Chao Lian, Shi-Qi Hu, Jin Zhang, Cai Cheng, Zhe Yuan, Shi-Wu Gao, and Sheng Meng. Integrated Plasmonics: Broadband Dirac Plasmons in Borophene. Phys. Rev. Lett. 2020, 125, 116802.”
“34. S. Verma, A. Mawrie, and T. K. Ghosh. Effect of electron-hole asymmetry on optical conductivity in 8-Pmmn borophene. Phys. Rev. B. 2017, 96, 155418.”
“35. M. A. Mojarro, R. Carrillo-Bastos, and J. A. Maytorena. Optical properties of massive anisotropic tilted Dirac systems. Phys. Rev. B. 2021, 103, 165415.”
“38. A. Singh, S. Ghosh, and A. Agarwal. Nonlinear and anisotropic polarization rotation in two-dimensional Dirac materials. Phys. Rev. B. 2018, 97, 205420.”
“40. Xing-Fei Zhou. Valley-dependent electron retroreflection and anomalous Klein tunneling in an 8-pmm borophene-based n − p − n junction. Phys. Rev. B. 2019, 100, 195139.”
“41. Xing-Fei Zhou. Valley splitting and anomalous Klein tunneling in borophane-based n − p and n − p − n junctions. Phys. Lett. A. 2020, 384, 126612.”
“42. Hong-Xia Zhong, Kai-Xiang Huang, Guo-Dong Yu and Sheng-Jun Yuan. Electronic and mechanical properties of few-layer borophene. Phys. Rev. B. 2018, 98, 054104.” “43. M. Nakhaee, S. A. Ketabi and F. M. Peeters. Dirac nodal line in bilayer borophene: Tight-binding model and low-energy effective Hamiltonian. Phys. Rev. B. 2018, 98, 115413.”
“44. A. E. Champo and G. G. Naumis. Metal-insulator transition in 8-Pmmn borophene under normal incidence of electromagnetic radiation. Phys. Rev. B. 2019, 99, 035415.”
“45. V. G. Ibarra-Sierra, J. C. Sandoval-Santana, A. Kunold and Gerardo G. Naumis. Dynamical band gap tuning in anisotropic tilted Dirac semimetals by intense elliptically polarized normal illumination and its application to 8-Pmmn borophene. Phys. Rev. B. 2019, 100, 125302.”
“46. G. C. Paul, SK F. Islam and A. Saha. Fingerprints of tilted Dirac cones on the RKKY exchange interaction in 8-Pmmn borophene. Phys. Rev. B. 2019, 99, 155418.”
“47. Shu-Hui Zhang, Ding-Fu Shao and Wen Yang. Velocity-determined anisotropic behaviors of RKKY interaction in 8-Pmmn borophene. J. Magn. Magn. Mater. 2019, 491, 165631.”
“48. Shu-Hui Zhang and Wen Yang. Velocity-determined anisotropic behaviors of RKKY interaction in 8-Pmmn borophene. New J. Phys. 2019, 21, 103052.”
“49. Miao Gao, Xun-Wang Yan, Jun Wang, Zhong-Yi Lu and Tao Xiang. Electron-phonon coupling in a honeycomb borophene grown on Al (111) surface. Phys. Rev. B. 2019, 100, 024503.”
“50. P. Kapri, B. Dey and T. K. Ghosh. Valley caloritronics in a photodriven heterojunction of Dirac materials. Phys. Rev. B. 2020, 102, 045417.”
“51. Jian-Long Zheng, Jun-Qiang Lu and Feng Zhai. Anisotropic and gate-tunable valley filtering based on 8-Pmmn borophene. Phys. Rev. B. 2020, 32, 025205.”
“52. Xing-Fei Zhou. Anomalous Andreev reflection in an 8-pmmn borophene-based superconducting junction. Phys. Rev. B. 2020, 102, 045132.”
at line 39-43.